# ReDS: Offline Reinforcement Learning With Heteroskedastic Datasets via Support Constraints

**Anikait Singh**[1,*], **Aviral Kumar**[1,*], **Quan Vuong**[2], **Yevgen Chebotar**[2], **Sergey Levine**[1]
[1]UC Berkeley, [2]Google DeepMind    (*Equal contribution)
asap7772@berkeley.edu

## Abstract

Offline reinforcement learning (RL) learns policies entirely from static datasets. Practical applications of offline RL will inevitably require learning from datasets where the variability of demonstrated behaviors changes non-uniformly across the state space. For example, at a red light, nearly all human drivers behave similarly by stopping, but when merging onto a highway, some drivers merge quickly, efficiently, and safely, while many hesitate or merge dangerously. Both theoretically and empirically, we show that typical offline RL methods, which are based on distribution constraints fail to learn from data with such non-uniform variability, due to the requirement to stay close to the behavior policy **to the same extent** across the state space. Ideally, the learned policy should be free to choose **per state** how closely to follow the behavior policy to maximize long-term return, as long as the learned policy stays within the support of the behavior policy. To instantiate this principle, we reweight the data distribution in conservative Q-learning (CQL) to obtain an approximate support constraint formulation. The reweighted distribution is a mixture of the current policy and an additional policy trained to mine poor actions that are likely under the behavior policy. Our method, CQL (ReDS), is theoretically motivated, and improves performance across a wide range of offline RL problems in games, navigation, and pixel-based manipulation.

## 1 Introduction

Recent advances in offline RL [39, 36] hint at exciting possibilities in learning high-performing policies, entirely from offline datasets, without requiring dangerous [19] or expensive [25] active interaction. Analogously to the importance of data diversity in supervised learning [9], the practical benefits of offline RL depend heavily on the *coverage* of behavior in the offline datasets [35]. Intuitively, the dataset must illustrate the consequences of a diverse range of behaviors, so that an offline RL method can determine what behaviors lead to high returns, ideally returns that are significantly higher than the best single behavior in the dataset.

One easy option to attain this kind of coverage is to combine many realistic sources of data, but doing so can lead to the variety of demonstrated behaviors varying in highly non-uniform ways across the state space, i.e. the dataset is *heteroskedastic*. For example, a driving dataset might show very high variability in driving habits, with some drivers being timid and some more aggressive, but remain remarkably consistent in "critical" states (e.g., human drivers are extremely unlikely to swerve in an empty road or drive off a bridge). A good offline RL algorithm should combine the *best* parts of each behavior in the dataset – e.g., in the above example, the algorithm should produce a policy that is *as good as the best human in each situation*, which would be better than *any* human driver overall. At the same time, the learned policy should not attempt to extrapolate to novel actions in subset of the state space where the distribution of demonstrated behaviors is narrow (e.g., the algorithm should not attempt to drive off a bridge). How effectively can current offline RL methods selectively choose on a *per-state* basis how closely to stick to the behavior policy?

Most existing methods [32, 33, 28, 27, 53, 17, 23] constrain the learned policy to stay close to the behavior policy with so-called "distribution constraints". Using a combination of empirical and theoretical evidence, we first show that distribution constraints are insufficient when the heteroskedasticity of the demonstrated behaviors varies non-uniformly across states, because the strength of the constraint is state-agnostic, and may be overly conservative at some states even when it is not conservative enough at other states. We also devise a measure of heteroskedasticity that enables us to determine if certain offline datasets would be challenging for distribution constraints.

Our second contribution is a simple observation: distribution constraints against a *reweighted* version of the behavior policy give rise to support constraints. That is, the return-maximization optimization process can freely choose per-state how much the learned policy should stay close to the behavior policy, so long as the learned policy remains within the data support. We show that it is convenient to instantiate this insight on top of conservative Q-learning (CQL) [33], a recent offline RL method. The new method, CQL (ReDS), changes minimally the form of regularization, design decisions employed by CQL and inherits existing hyper-parameter values. CQL (ReDS) attains better performance than recent distribution constraints methods on a variety of tasks with more heteroskedastic distributions.

## 2 Preliminaries

The goal in offline RL is find the optimal policy in a Markov decision process (MDP) specified by the tuple $\mathcal{M} = (\mathcal{S}, \mathcal{A}, T, r, \mu_0, \gamma)$. $\mathcal{S}, \mathcal{A}$ denote the state and action spaces. $T(\mathbf{s}'|\mathbf{s}, \mathbf{a})$ and $r(\mathbf{s}, \mathbf{a})$ represent the dynamics and reward function. $\mu_0(s)$ denotes the initial state distribution. $\gamma \in (0, 1)$ denotes the discount factor. We wish to learn a policy that maximizes return, denoted by $J(\pi) := \frac{1}{1-\gamma} \mathbb{E}_{(\mathbf{s}_t, \mathbf{a}_t) \sim \pi}[\sum_t \gamma^t r(\mathbf{s}_t, \mathbf{a}_t)]$. We must find this policy while only having access to an offline dataset of transitions collected using a behavior policy $\pi_\beta$, $\mathcal{D} = \{(\mathbf{s}, \mathbf{a}, r, \mathbf{s}')\}$.

**Offline RL via distributional constraints**. Most offline RL algorithms regularize the learned policy $\pi$ from querying the target Q-function on unseen actions [17, 30], either implicitly or explicitly. For our theoretical analysis, we will abstract the behavior of distributional constraint offline RL algorithms into a generic formulation following Kumar et al. [33]. As shown in Equation 1, we consider the problem where we must maximize the return of the learned policy $\pi$ (in the empirical MDP) $\widehat{J}(\pi)$, while also penalizing the divergence from $\pi_\beta$:

$$\max_\pi \ \mathbb{E}_{\mathbf{s} \sim \widehat{d}^\pi} \left[ \widehat{J}(\pi) - \alpha D(\pi, \pi_\beta)(\mathbf{s}) \right], \tag{1}$$

where $D$ denotes a divergence between the learned policy $\pi$ and the behavior policy $\pi_\beta$ at state $\mathbf{s}$.

**Conservative Q-learning.** [33] enforces the distributional constraint on the policy *implicitly*. To see why this is the case, consider the CQL objective, which consists of two terms:

$$\min_\theta \ \alpha \underbrace{(\mathbb{E}_{\mathbf{s} \sim \mathcal{D}, \mathbf{a} \sim \pi}[Q_\theta(\mathbf{s}, \mathbf{a})] - \mathbb{E}_{\mathbf{s}, \mathbf{a} \sim \mathcal{D}}[Q_\theta(\mathbf{s}, \mathbf{a})])}_{\mathcal{R}(\theta)} + \frac{1}{2} \mathbb{E}_{\mathbf{s}, \mathbf{a}, \mathbf{s}' \sim \mathcal{D}} \left[ \left( Q_\theta(\mathbf{s}, \mathbf{a}) - \mathcal{B}^\pi \bar{Q}(\mathbf{s}, \mathbf{a}) \right)^2 \right], \tag{2}$$

where $\mathcal{B}^\pi \bar{Q}(\mathbf{s}, \mathbf{a})$ is the Bellman backup operator applied to a delayed target Q-network, $\bar{Q}$: $\mathcal{B}^\pi \bar{Q}(\mathbf{s}, \mathbf{a}) := r(\mathbf{s}, \mathbf{a}) + \gamma \mathbb{E}_{\mathbf{a}' \sim \pi(\mathbf{a}'|\mathbf{s}')}[\bar{Q}(\mathbf{s}', \mathbf{a}')]$. The second term (in blue) is the standard TD error [40, 18, 22]. The first term $\mathcal{R}(\theta)$ (in red) attempts to prevent overestimation in the Q-values for out-of-distribution (OOD) actions by minimizing the Q-values under a distribution $\mu(\mathbf{a}|\mathbf{s})$, which is automatically chosen to pick actions with high Q-values $Q_\theta(\mathbf{s}, \mathbf{a})$, and counterbalances by maximizing the Q-values of the actions in the dataset. Kumar et al. [33] show that Equation 2 gives rise to a pessimistic Q-function that modifies the optimal Q function by the ratios of densities, $\pi(\mathbf{a}|\mathbf{s})/\pi_\beta(\mathbf{a}|\mathbf{s})$ at a given state-action pair $(\mathbf{s}, \mathbf{a})$. Formally, the Q-function obtained after one iteration is given by:

$$Q_\theta(\mathbf{s}, \mathbf{a}) := \mathcal{B}^\pi \bar{Q}(\mathbf{s}, \mathbf{a}) - \alpha \left[ \frac{\pi(\mathbf{a}|\mathbf{s})}{\pi_\beta(\mathbf{a}|\mathbf{s})} - 1 \right]. \tag{3}$$

The Q function is unchanged only if the density of the learned policy $\pi$ matches that of the behavior policy $\pi_\beta$. Otherwise, for state-action pairs where $\pi(\mathbf{a}|\mathbf{s}) < \pi_\beta(\mathbf{a}|\mathbf{s})$, Eq. 3 increases their Q values and encourages the policy $\pi$ to assign more mass to the action. Vice versa, if $\pi(\mathbf{a}|\mathbf{s}) > \pi_\beta(\mathbf{a}|\mathbf{s})$, Eq. 3 encourages the policy $\pi$ to assign smaller density to the action $\mathbf{a}$. In Eq. 3, $\alpha$ is a constant for every state, and hence the value function learned by CQL is altered by the ratio of action probabilities to the same extent at all possible state-action pairs. As we will discuss in the next section, this can be sub-optimal when the learnt policy should stay close to the behavior policy in some states, but not others. We elaborate on this intuition in the next section.

# 3 Why Distribution Constraints Fail with Heteroskedastic Data

In statistics, heteroskedasticity is typically used to refer to the condition when the standard deviation in a given random variable varies non-uniformly over time (see for example, Cao et al. [6]). We call a offline dataset heteroskedastic when the variability of the behavior differs in different regions of the state space: for instance, if for certain regions of the state space, the observed behaviors in the dataset assign the most probability mass to a few actions, but in other regions, the observed behaviors are more diverse. Realistic offline datasets are often heteroskedastic as they are typically generated by multiple policies, each with its own characteristics, under different conditions. E.g., driving datasets come from multiple humans [11], and many robotic datasets are collected by multiple teleoperators [10], resulting in systematic variability in different regions of the state space.

## 3.1 A Didactic Example

To understand why distribution constraints are insufficient with heteroskedastic data, we present a didactic example. Motivated by the driving scenario, we consider a maze navigation task shown in Fig. 1. The task is to navigate from the position labeled as "Start" to the position labeled as "Goal" using five actions at every possible state (L: ←, R: →, U: ↑, D: ↓, No: No Op), while making sure that the executed actions do not hit the walls of the grid.

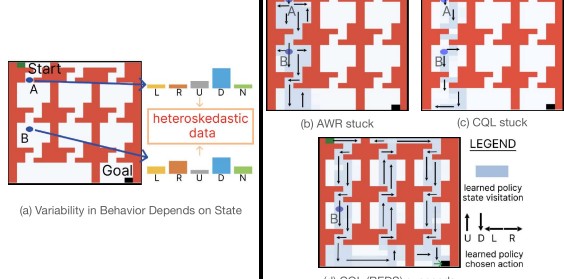

Figure 1: **Failure mode of distribution constraints.** In this navigation task, an offline RL algorithm must find a path from the start state to the goal state as indicated in (a). The offline dataset provided exhibits non-uniform coverage at different states, e.g., in the state marked as "B" located in a wide room has more uniform action distribution, whereas the states in the narrow hallways exhibit a more narrow action distribution. This is akin to how the behavior of human drivers varies in certain locations ("B"), but is very similar in other situations ("A"). To perform well, an algorithm must stay close to the data in the hallways ("A"), but deviate significantly from the data in the rooms ("B"), where the data supports many different behaviors (most are not good). AWR and CQL get stuck because they stay too close to the bad behavior policy in the rooms, e.g. the left and right arrows near State B in Fig (b) and (c). Our method, CQL (ReDS), learns to ignore the bad behavior action in state B and prioritizes the good action, indicated by the downward arrow near B in (d).

**Dataset construction.** To collect a heteroskedastic dataset, we consider a mixture of several behavior policies that attain a uniform occupancy over different states in the maze. However, the dataset action distributions differ significantly in different states. The induced action distribution is heavily biased to move towards the goal in the narrow hallways (e.g., the behavior policy moves upwards at state A)). In contrast, the action distribution is quite diverse in the wider rooms. In these rooms, the behavior policy often selects actions that do not immediately move the agent towards the goal (e.g., the behavior policy at state B), because doing so does not generally hit the walls as the rooms are wider, and hence the agent is not penalized. Whereas, the agent must take utmost precaution to not hit the walls in the narrow hallways. More details are in Appendix B.

**Representative distribution constraint algorithms** such as AWR [44, 43] and CQL [33] fail to perform the task, as shown in Figure 1. To ensure fair comparison, we tune each method to its best evaluation performance using online rollouts. The visualization in Figure 1 demonstrates that these two algorithms fail to learn reasonable policies because the learned policies match the random behavior of the dataset actions too closely in the wider rooms, and therefore are unable to make progress towards the Goal position. This is a direct consequence of enforcing too strong of a constraint on the learned policy to stay close to the behaviors in the dataset. Therefore, we also evaluated the performance of CQL and AWR in this example, with lower amounts of conservatism (Appendix B) and found that utilizing a lower amount of conservatism suffers from the opposite failure mode: it is unable to prevent the policies from hitting the walls in the narrow hallways. This means that conservatism prevents the algorithm from making progress in the regions where the behavior in the dataset is more diverse, whereas not being conservative enough hurts performance in regions where the behaviors in the dataset agree with each other. The method we propose in this paper to tackle this challenge, indicated as "CQL (ReDS)", effectively traverses the maze, 80% of the time.

## 3.2 Challenges with Distribution Constraints

Having seen that distribution constraints can fail in certain scenarios, we now formally characterize when offline RL datasets is heteroskedastic, and why distribution constraints may be ineffective in

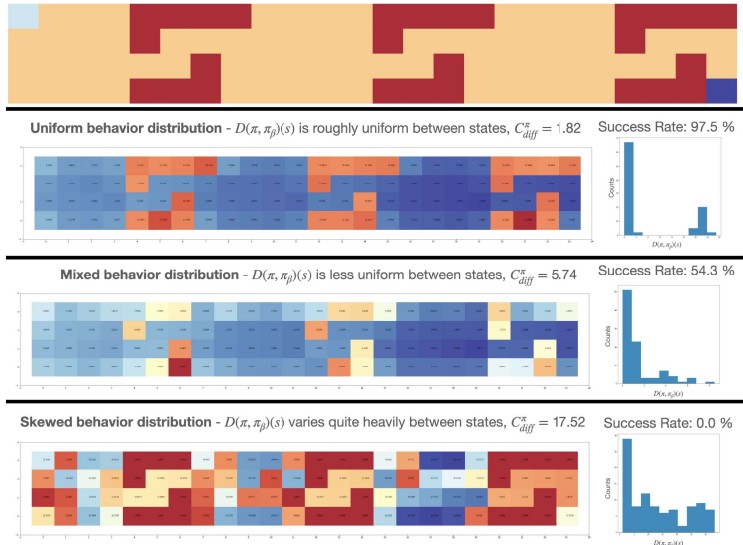

Figure 2: **Empirically computing** $C_{\text{diff}}^{\pi}$ with three datasets: uniform (top), mixed (middle) and skewed (bottom) on a gridworld. We also visualize $D(\pi, \pi_\beta)(\mathbf{s})$ across states in the maze as the colors on different cells, a histogram of $D(\pi, \pi_\beta)(\mathbf{s})$ to visualize variation in this quantity and the performance of running standard CQL. **Top:** The uniform distribution leads to low $C_{\text{diff}}^{\pi}$, uniform $D(\pi, \pi_\beta)(s)$, and highest success. **Middle:** The mixed distribution leads to medium $C_{\text{diff}}^{\pi}$, less uniformly distributed $D(\pi, \pi_\beta)(s)$, and a drop in task success. **Bottom**: The skewed distribution leads to a high $C_{\text{diff}}^{\pi}$, non-uniform $D(\pi, \pi_\beta)(s)$, and poor performance.

such scenarios. Similar to how standard analyses utilize concentrability coefficient [46], which upper bounds the ratio of state-action visitation under a policy $d^{\pi}(\mathbf{s}, \mathbf{a})$ and the dataset distribution $\mu$, i.e., $\max_{\mathbf{s},\mathbf{a}} d^{\pi}(\mathbf{s}, \mathbf{a})/\mu(\mathbf{s}, \mathbf{a}) \leq C^{\pi}$, we introduce a new metric called *differential concentrability*, which measures dataset heteroskedasticity (i.e., the variability in the dataset behavior across different states).

**Definition 3.1** (Differential concentrability.). Given a divergence $D$ over the action space, the differential concentrability of a given policy $\pi$ with respect to the behavioral policy $\pi_\beta$ is given by:

$$C_{\text{diff}}^{\pi} = \mathop{\mathbb{E}}_{\mathbf{s}_1, \mathbf{s}_2 \sim d^{\pi}} \left[ \left( \sqrt{\frac{D(\pi, \pi_\beta)(\mathbf{s}_1)}{\mu(\mathbf{s}_1)}} - \sqrt{\frac{D(\pi, \pi_\beta)(\mathbf{s}_2)}{\mu(\mathbf{s}_2)}} \right)^2 \right]. \tag{4}$$

Eq. 4 measures the variation in the divergence between a given policy $\pi(\mathbf{a}|\mathbf{s})$ and the behavior policy $\pi_\beta(\mathbf{a}|\mathbf{s})$ weighted inversely by the density of these states in the offline dataset (i.e., $\mu(\mathbf{s})$ in the denominator). For simplicity, let us revisit the navigation example from Section 3.1 and first consider a scenario where $\mu(\mathbf{s}) = \text{Unif}(\mathcal{S})$. For any given policy $\pi$, if there are states where $\pi$ chooses actions that lie on the fringe of the data distribution (e.g., in the wider rooms), as well as states where the policy $\pi$ chooses actions at the mode of the data distribution (e.g., as in the narrow passages), then $C_{\text{diff}}^{\pi}$ would be large any policy $\pi$ that we learn. Crucially, $C_{\text{diff}}^{\pi}$ would be small even if the learned policy $\pi$ deviates significantly from the behavior policy $\pi_\beta$, such that $D(\pi, \pi_\beta)(\mathbf{s})$ is large, but $|D(\pi, \pi_\beta)(\mathbf{s}_1) - D(\pi, \pi_\beta)(\mathbf{s}_2)|$ is small, indicating the dataset is not heteroskedastic.

**Connection between variability in the action distribution and high $C_{\text{diff}}^{\pi}$.** Consider a simpler formula where we remove the counts $n(\mathbf{s})$ from the expression of differential concentrability and set $\pi$ in $C_{\text{diff}}^{\pi}$ to be the uniform distribution over actions. Then, we can show that the value of $C_{\text{diff}}^{\pi}$ is *exactly* equal to twice the variance of $D(\pi, \pi_\beta)(\mathbf{s})$ across states. Therefore, we will demonstrate in Section 5 that arbitrary policy checkpoints $\pi$ learned by offline RL algorithms generally attain a low value of the variance in $D(\pi, \pi_\beta)(\mathbf{s})$ on offline datasets from non-heteroskedastic sources, such as those covered in the D4RL [13] benchmark. Of course, we cannot always exclude the counts of states $n(\mathbf{s})$, however, we note that in high-dimensional state spaces, such as those in our experiments, each state in the offline data is likely to be unique, thus validating the condition that $n(\mathbf{s}) = 1$. That said, we do compute the exact value of $C_{\text{diff}}^{\pi}$ (with $n(\mathbf{s})$) in a didactic gridworld maze shown in Figure 2. In this case, we find that our definition of $C_{\text{diff}}^{\pi}$ is actually able to reflect the intuitive notion of heteroskedasticity.

We now use the definition of differential concentrability to bound both the improvement and deprovement of $\pi$ w.r.t. $\pi_\beta$ for distribution constraint algorithms using the framework of safe policy

improvement [37, 33]. We show that when $C_{\text{diff}}^{\pi}$ is large, then constraints (Eq. 1) may not improve significantly over $\pi_\beta$, even for the best value for the weight $\alpha$ (proof in Appendix C):

**Theorem 3.2** (Informal; Limited policy improvement via distributional constraints.). *W.h.p.* $\geq 1 - \delta$, *for any prescribed level of safety $\zeta$, the maximum possible policy improvement over choices of $\alpha$,* $\max_\alpha \ [J(\pi_\alpha) - J(\pi_\beta)] \leq \zeta^+$, *where $\zeta^+$ is given by:*

$$\zeta^+ := \max_\alpha \quad \frac{h^*(\alpha)}{(1-\gamma)^2} s.t. \quad \frac{c_1 \sqrt{\log \frac{|\mathcal{S}||\mathcal{A}|}{\delta}}}{(1-\gamma)^2} \frac{\sqrt{C_{\text{diff}}^{\pi_\alpha}}}{|\mathcal{D}|} - \frac{\alpha \mathbb{E}_{\mathbf{s} \sim \widehat{d}^{\pi_\alpha}}[D(\pi_\alpha, \pi_\beta)(\mathbf{s})]}{1 - \gamma} \leq \zeta, \qquad (5)$$

*where $h^*$ is a monotonically decreasing function of $\alpha$, and $h(0) = \mathcal{O}(1)$.*

Theorem 3.2 quantifies the fundamental tradeoff with distribution constraints: to satisfy a given $\zeta$-safety constraint in problems with larger $C_{\text{diff}}^{\pi}$, we would need a larger $\alpha$. Since the maximum policy improvement $\zeta^+$ is upper bounded by $h^*(\alpha)$, the policy may not necessarily improve over the behavior policy if $\alpha$ is large. On the flip side, if we choose to fix the value of $\alpha$ to be small in hopes to attain more improvement in problems where $C_{\text{diff}}^{\pi}$ is high for all policies, we would end up compromising on the safety guarantee as $\zeta$ needs to be large for a small $\alpha$ and large $C_{\text{diff}}^{\pi}$. Thus, in this case, the policy may not improve over the behavior policy reliably.

Note that a larger value of $C_{\text{diff}}^{\pi}$ need not imply large $\mathbb{E}_{\mathbf{s} \sim \widehat{d}^{\pi}}[D(\pi, \pi_\beta)(\mathbf{s})]$ because the latter does not involve $\mu(\mathbf{s})$. $C_{\text{diff}}^{\pi}$ also measures the dispersion of $D(\pi, \pi_\beta)(\mathbf{s})$, while the latter performs a mean over states. In addition, Theorem 3.2 characterizes the *maximum possible* improvement with an *oracle* selection of $\alpha$, though is not feasible in practice. Thus, when $C_{\text{diff}}^{\pi}$ is large, distribution constraint algorithms could either not safely improve over $\pi_\beta$ or would attain only a limited improvement with *any possible* value of $\alpha$. Finally, we remark that complementing [32, 39] that discuss failure modes of distribution constraints with high-entropy behavior policies, Theorem 3.2 quantifies when this would be the case: this happens when $C_{\text{diff}}^{\pi}$ is large.

# 4   Support Constraints As Reweighted Distribution Constraints

Thus far, we have seen that distribution constraints can be ineffective with heteroskedastic datasets. If we can impose the distribution constraint such that the constraint strength can be modulated per state, then in principle, we can alleviate the issue raised in Theorem 3.2 and Section 3.1.

**Our key insight** is that by reweighting the action distribution in the data before utilizing a distribution constraint, we can obtain a method that enforces a per-state distribution constraint, which corresponds to an approximate *support* constraint. This will push down the values of actions that are outside the behavior policy support, but otherwise not impose a severe penalty for in-support actions, thus enabling the policy to deviate from the behavior policy by different amounts at different states. Rather than having a distribution constraint between $\pi$ and $\pi_\beta$ (Eq. 1), if we can impose a constraint between $\pi$ and a *reweighted* version of $\pi_\beta$, where the reweighting is state-dependent, then we can obtain an approximate support constraint. Let the reweighted distribution be $\pi^{re}$. Intuitively, if $\pi(\cdot|\mathbf{s})$ is within the support of the $\pi_\beta(\cdot|\mathbf{s})$, then one can find a reweighting $\pi^{re}(\cdot|\mathbf{s})$ such that $D(\pi, \pi^{re})(\mathbf{s}) = 0$, whereas if $\pi(\cdot|\mathbf{s})$ is not within the support of $\pi^{re}(\cdot|\mathbf{s})$, then $D(\pi, \pi^{re})(\mathbf{s})$ still penalizes $\pi$ when $\pi$ chooses out-of-support actions, since no reweighting $\pi^{re}$ can put non-zero probability on out-of-support actions. This allows us to handle the failure mode from Section 3: at states with wide behavior policy, even with a large $\alpha$, $\pi$ is not anymore constrained to the behavior distribution, whereas at other "critical" states, where $\pi_\beta$ is narrow, a large enough $\alpha$ will constrain $\pi(\cdot|\mathbf{s})$ to stay close to $\pi_\beta(\cdot|\mathbf{s})$. We call this **Re**weighting **D**istribution constraints to **S**upport (ReDS).

## 4.1   Instantiating the Principle Behind ReDS

One option is to reweight $\pi_\beta$ to $\pi^{re}$, and enforce a distribution constraint $D(\pi, \pi^{re})$ between $\pi$ and $\pi^{re}$. However, this is problematic because the $\pi^{re}$ would typically be estimated by using importance weighting or by fitting a parametric model, and prior work has shown that errors in estimating the behavior policy [43, 20] using only one action sample often get propagated and lead to poor downstream performance. For CQL, this issue might be especially severe if we push up the Q-values under $\pi^{re}$, because then these errors might lead to severe Q-value over-estimation.

**Abstract idea of CQL (ReDS).** Instead, we devise an alternative formulation for ReDS that modifies the learned policy $\pi$ to $\pi^{re}$, such that applying a distribution constraint on this modified policy imposes a support constraint. Thus, with CQL, now we instead *push down* the Q-values under $\pi^{re}$. We define $\pi^{re}$ as a mixture distribution of the learned policy $\pi$ and a reweighted version of the behavior policy as follows:

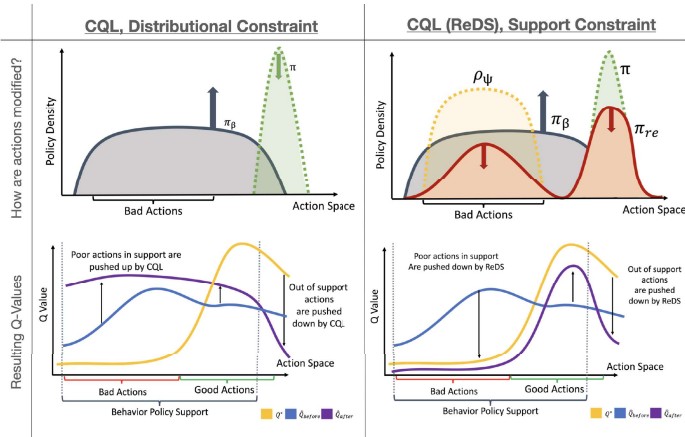

Figure 3: **Comparison between support and distributional constraints: Left:** CQL pushes down the Q-function under the policy $\pi$, while pushing up the function under the behavior policy $\pi_\beta$. This means that the Q-values for bad actions can go up. **Right:** In contrast, ReDS re-weights the data distribution to push down the values of bad actions, alleviating this shortcoming.

$$\pi^{re}(\cdot|\mathbf{s}) := \frac{1}{2}\pi(\cdot|\mathbf{s}) + \frac{1}{2}\left[\pi_\beta(\cdot|\mathbf{s}) \cdot g\left(\pi(\cdot|\mathbf{s})\right)\right], \tag{6}$$

where $g(\cdot)$ is a monotonically decreasing function. We will demonstrate how pushing down the Q-values under $\pi^{re}$ modifies CQL to enable a support constraint while reusing existing components of CQL that impose a distribution constraint. As shown in Figure 3, the second term in Equation 6 increases the probability of actions that are likely under the behavior policy, but are less likely under the learned policy (due to $g$ being a decreasing function). We will show in Lemma 4.1 that utilizing $\pi^{re}$ in CQL enforces a support constraint on $\pi$. Thus, the learned policy $\pi$ can be further away from $\pi_\beta$, allowing $\pi$ to assign more probability to good actions that are within the behavior policy support, even if they have lower probabilities under $\pi_\beta$. Section 4.2 illustrates theoretically why pushing down the Q-values under Eq. 6 approximates a support constraint in terms of how it modifies the resulting Q-values. For an illustration, please see Figure 3.

**How should we pick $g$ in practice?** Since we wish to use $\pi^{re}$ as a replacement for $\pi$ in the minimization term in the CQL regularizer (Equation 2), we aim to understand how to design the re-weighting $g$ in practice. Since specifically CQL enforces a distribution constraint by maximizing the Q-value on *all* actions sampled from the behavior policy $\pi_\beta$, our choice of $g$ should aim to counter this effect by instead minimizing the Q-value on "bad" actions within the support of the behavior policy. Equation 6 quantifies the notion of these "bad" actions using a monontonically decreasing function $g(\pi(\mathbf{a}|\mathbf{s}))$ of the policy probability. In practice, we find it convenient to define $g$ to be a function of the advantage estimate: $A_\theta(\mathbf{s}, \mathbf{a}) := Q_\theta(\mathbf{s}, \mathbf{a}) - E_{\mathbf{a}\sim\pi}[Q_\theta(\mathbf{s}, \mathbf{a})]$, that the policy $\pi$ is seeking to maximize. In fact, if entropy regularization is utilized for training the policy (akin to most offline RL algorithms), the density of an action under a policy is directly proportional to exponentiated advantages, i.e., $\pi(\mathbf{a}|\mathbf{s}) \propto \exp(A_\theta(\mathbf{s}, \mathbf{a}))$. Hence, we choose $g(x) = 1/x$, such that $g(\exp(A(\mathbf{s}, \mathbf{a}))) = \exp(-A(\mathbf{s}, \mathbf{a}))$ (a decreasing function).

For the rest, we approximate the product distribution $\pi(\mathbf{a}|\mathbf{s}) \cdot g(\pi(\mathbf{a}|\mathbf{s}))$ by fitting a parametric function approximator $\rho_\psi(\mathbf{a}|\mathbf{s})$. Since $\rho_\psi(\mathbf{a}|\mathbf{s})$ is being trained to approximate a re-weighted version of the behavior policy, we fit $\rho_\psi$ by minimizing using a weighted maximum log-likelihood objective, as shown in prior work [44, 43]. The concrete form for our objective for training $\rho_\psi$ is shown below ($\tau$ is a temperature hyperparameter typically introduced in prior work [44, 43]):

$$\rho_\psi(\cdot|\mathbf{s}) = \arg\max_{\rho_\psi} \ \mathbb{E}_{\mathbf{s}\sim\mathcal{D}, \mathbf{a}\sim\pi_\beta(\cdot|\mathbf{s})}[\log \rho_\psi(\mathbf{a}|\mathbf{s}) \cdot \exp\left(-A_\theta(\mathbf{s}, \mathbf{a})/\tau\right)]. \tag{7}$$

The crucial difference between this objective and standard advantage-weighted updates is the difference of the sign. While algorithms such as AWR [43] aim to find an action that attains a high advantage while being close to the behavior policy, and hence, uses a positive advantage, we utilize the *negative* advantage to mine for poor actions that are still quite likely under the behavior policy.

The final objective for the Q-function combines the regularizer in Eq. 8 with a standard TD objective:

$$\mathcal{R}(\theta; {\color{red}\rho}) = \frac{1}{2}\left(\mathop{\mathbb{E}}_{\mathbf{s}\sim\mathcal{D},\mathbf{a}\sim\pi}[Q_\theta(\mathbf{s},\mathbf{a})] + \mathop{\mathbb{E}}_{\mathbf{s}\sim\mathcal{D},\mathbf{a}\sim\rho}[Q_\theta(\mathbf{s},\mathbf{a})]\right) - \mathop{\mathbb{E}}_{\mathbf{s},\mathbf{a}\sim\mathcal{D}}[Q_\theta(\mathbf{s},\mathbf{a})] \tag{8}$$

$$\min_\theta \; J_Q(\theta) = {\color{red}\mathcal{R}(\theta;\rho)} + \frac{1}{2}\mathbb{E}_{\mathbf{s},\mathbf{a},\mathbf{s}'\sim\mathcal{D}}[(Q_\theta(\mathbf{s},\mathbf{a}) - \mathcal{B}^\pi\bar{Q}(\mathbf{s},\mathbf{a}))^2] \tag{9}$$

## 4.2  Theoretical Analysis of CQL (ReDS)

Next, we analyze CQL (ReDS), showing how learning using the regularizer in Eq. 8 modifies the Q-values and justifies our choice of the distribution $\rho$ in the previous section.

**Lemma 4.1** (Per-state change of Q-values.). *Let $g(\mathbf{a}|\mathbf{s})$ be a shorthand for $g(\mathbf{a}|\mathbf{s}) = g(\tau \cdot \pi(\mathbf{a}|\mathbf{s}))$. In the tabular setting, the Q-function obtained after one iteration of objective in Eq. 9 is given by:*

$$Q_\theta(\mathbf{s},\mathbf{a}) := \mathcal{B}^\pi\bar{Q}(\mathbf{s},\mathbf{a}) - \alpha\frac{\pi(\mathbf{a}|\mathbf{s}) + \pi_\beta(\mathbf{a}|\mathbf{s})g(\mathbf{a}|\mathbf{s}) - 2\pi_\beta(\mathbf{a}|\mathbf{s})}{2\pi_\beta(\mathbf{a}|\mathbf{s})} \tag{10}$$

*where $\mathcal{B}^\pi\bar{Q}(\mathbf{s},\mathbf{a})$ is the Bellman backup operator applied to a delayed target Q-network.*

Eq. 10 illustrates why the modified regularizer in Eq. 8 leads to a "soft" support constraint whose strength is modulated per-state. Since $g$ is a monotonically decreasing function of $\pi$, for state-action pairs where $\pi(\mathbf{a}|\mathbf{s})$ has high values, $g(\mathbf{a}|\mathbf{s})$ is low and therefore the Q-value $Q(\mathbf{s},\mathbf{a})$ for such state-action pairs are underestimated less. Vice versa, for state-action pairs where $\pi(\mathbf{a}|\mathbf{s})$ attains low values, $g(\mathbf{a}|\mathbf{s})$ is high to counter-acts the low $\pi(\mathbf{a}|\mathbf{s})$ values. Also, since $\pi_\beta(\mathbf{a}|\mathbf{s})$ appears in the denominator, for out-of-support actions, where $\pi_\beta(\mathbf{a}|\mathbf{s}) = 0$, $\pi(\mathbf{a}|\mathbf{s})$ must also assign $0$ probability to the actions for the Q values to be well defined. An illustration of this idea is shown in Figure 9. We can use this insight to further derive the closed-form objective optimized by ReDS.

**Lemma 4.2** (CQL (ReDS) objective.). *Assume that for all policies $\pi \in \Pi, \forall(\mathbf{s},\mathbf{a}), \pi(\mathbf{a}|\mathbf{s}) > 0$. Then, CQL (ReDS) solves the following optimization problem:*

$$\max_{\pi\in\Pi} \; \widehat{J}(\pi) - \frac{\alpha}{2(1-\gamma)}\mathbb{E}_{\mathbf{s}\sim\widehat{d}^\pi}\left[D(\pi,\pi_\beta)(\mathbf{s}) + \mathop{\mathbb{E}}_{\mathbf{a}\sim\pi(\cdot|\mathbf{s})}[g(\tau\cdot\pi(\mathbf{a}|\mathbf{s}))\,\mathbb{I}\{\pi_\beta(\mathbf{a}|\mathbf{s}) > 0\}]\right]. \tag{11}$$

$\widehat{J}(\pi)$ corresponds to the empirical return of the learned policy, i.e., the return of the policy under the learned Q-function. The objective in Lemma 4.3 can be intuitively interpreted as follows: The first term, $D(\pi,\pi_\beta)(\mathbf{s})$, is a standard distribution constraint, also present in naïve CQL, and it aims to penalize the learned policy $\pi$ if it deviates too far away from $\pi_\beta$. ReDS adds an additional second term that effectively encourages $\pi$ to be "sharp" within the support of the behavior policy (as $g$ is monotonically decreasing), enabling $\pi$ to potentially put mass on actions that lead to a high $\widehat{J}(\pi)$.

Specifically, this second term allows us control the strength of the distribution constraint per state: at states where the support of the policy is narrow, i.e., the volume of actions such that $\pi_\beta(\mathbf{a}|\mathbf{s}) > 0$ is small (say, only a single action), the penalty in Equation 51 reverts to a standard distributional constraint by penalizing divergence from the behavioral policy via $D(\pi,\pi_\beta)(\mathbf{s})$ as the second term cannot be minimized. At states where the policy $\pi_\beta$ is broad, the second term counteracts the effect of the distributional constraint within the support of the behavior policy, by enabling $\pi$ to concentrate its density on only good actions within the support of $\pi_\beta$ with the same multiplier $\alpha$. Thus even when we need to set $\alpha$ to be large to stay close to $\pi_\beta(\cdot|\mathbf{s})$ at certain states (e.g., in narrow hallways in the example in Sec. 3.1), $D(\pi,\pi_\beta)(\mathbf{s})$ is not heavily constrained at other states.

In fact, we formalize this intuition below to show that for the best possible value of the hyperparameters appearing in the training objective for CQL (ReDS) (Equation 51), CQL (ReDS) is guaranteed to outperform the best-tuned version of CQL for any offline RL problem. A proof is in Appendix C.

**Lemma 4.3** (CQL (ReDS) formal guarantee). *We will add the following guarantee to show that the policy learned by ReDS for the best possible value of $\tau$ (Equation 51) and $\alpha$ in CQL (Equation 3) outperforms the best CQL policy. That is, formally we show:*

$$\max_{\alpha,\tau} J(\pi_{\mathrm{ReDS};\alpha,\tau}) \geq \max_\alpha J(\pi_{\mathrm{CQL};\alpha}). \tag{12}$$

## 5  Experimental Evaluation

The goal of our experiments is to understand how CQL (ReDS) compares to distributional constraint methods when learning from heteroskedastic offline datasets. In order to perform our experiments,

we construct new heteroskedastic datasets that pose challenges representative of what we would expect to see in real-world problems. We first introduce tasks and heteroskedastic datasets that we evaluate on, and then present our results compared to prior state-of-the-art methods. We also evaluate ReDS on some of the standard D4RL [13] datasets which are not heteroskedastic in and find that the addition of ReDS, as expected, does not help, or hurt on those tasks.

## 5.1 Comparison on the D4RL Benchmark

| Dataset | BC | 10%BC | DT | AWAC | Onestep RL | TD3+BC | COMBO | CQL | IQL | **Ours** |
|---|---|---|---|---|---|---|---|---|---|---|
| halfcheetah-medium-replay | 36.6 | 40.6 | 36.6 | 40.5 | 38.1 | 44.6 | **55.1** | 45.5 | 44.2 | **52.3** |
| hopper-medium-replay | 18.1 | 75.9 | 82.7 | 37.2 | **97.5** | 60.9 | 89.5 | 95.0 | 94.7 | **101.5** |
| walker2d-medium-replay | 26.0 | 62.5 | 66.6 | 27.0 | 49.5 | **81.8** | 56.0 | 77.2 | 73.9 | **85.0** |
| halfcheetah-medium-expert | 55.2 | 92.9 | 86.8 | 42.8 | **93.4** | 90.7 | 90.0 | **91.6** | 86.7 | 89.5 |
| hopper-medium-expert | 52.5 | 110.9 | 107.6 | 55.8 | 103.3 | 98.0 | **111.1** | 105.4 | 91.5 | **110.0** |
| walker2d-medium-expert | 107.5 | 109.0 | 108.1 | 74.5 | **113.0** | 110.1 | 103.3 | 108.8 | 109.6 | **112.0** |
| locomotion total | 295.9 | 491.8 | 488.4 | 277.8 | 494.8 | 486.1 | 505 | **523.5** | 500.6 | **550.3** |

Table 1: Performance comparison on the D4RL benchmark. (Top 2 **bolded**)

Heteroskedastic data is likely to exist in real-world problems such as driving and manipulation, where datasets are collected by multiple policies that agree and disagree at different states. While standard benchmarks (D4RL [13] and RLUnplugged [21]) include offline datasets generated by mixture policies (e.g. the "medium-expert" generated by two policies with different performance), these policies are trained via RL methods (SAC) that constrain the entropy of the action distribution at each state to be uniform. To measure heteroskedasticity, we utilize an approximation to $C_{\text{diff}}^{\pi}$: the standard deviation in the value of $D(\pi, \pi_\beta)(\mathbf{s})$ across states in the dataset, using a fixed policy $\pi$ obtained by running CQL. We didn't use $C_{\text{diff}}^{\pi}$ directly, as it is challenging to compute in continuous spaces. In Table 3, the standard deviation is lower for the D4RL antmaze datasets, corroborating our intuition that these datasets are significantly less heteroskedastic.

## 5.2 Comparisons on Heteroskedastic datasets

**Heteroskedastic datasets.** To stress-test our method and prior distribution constraint approaches, we collected new datasets for the medium and large mazes used in the antmaze navigation tasks from D4RL: `noisy` datasets, where the behavior policy action variance differs in different regions of the maze, representative of user variability in navigation, and `biased` datasets, where the behavior policy admits a systematic bias towards certain behaviors in different regions of the maze, representative of bias towards certain routes in navigation problems. Table 3 shows that these datasets are significantly more heteroskedastic to the D4RL datasets.

| Dataset | std | max |
|---|---|---|
| noisy (Ours) | **18** | **253** |
| biased (Ours) | **9** | **31** |
| diverse (D4RL) | 2 | 11 |
| play (D4RL) | 2 | 13 |

Table 3: The new antmaze datasets (Ours) are significantly more heteroskedastic than the standard D4RL datasets. We measure heteroskedasticity using the std and max of $D(\pi, \pi_\beta)(\mathbf{s})$ across states in the offline dataset.

Using these more heteroskedastic datasets, we compare CQL (ReDS) with CQL and IQL [27], recent popular methods, and two prior methods, BEAR [30] and EDAC [3], that also enforce support constraints. For each algorithm, including ours, we utilize hyperparameters directly from the counterpart tasks in D4RL. Due to the lack of an effective method for offline policy selection (see Fu et al. [14]), we utilize oracle checkpoint selection for every method. We compute the mean and standard deviation across 3 seeds. Table 2 shows that the largest gap between CQL (ReDS) and prior methods is on `noisy` datasets, which are particularly more heteroskedastic (Table 3).

We also compare CQL (ReDS) with recent offline RL algorithms on D4RL, including DT [8], AWAC [42], onestep RL [5], TD3+BC [16] and COMBO [57]. Table 1 shows that CQL (ReDS) obtains similar performance as existing distributional constraint methods and outperforms BC-based baselines. This is expected given that the D4RL datasets exhibit significantly smaller heteroscedasticity, as previously explained. Also, a large fraction of the datasets is trajectories with high returns. BC using the top 10% trajectories with the highest episode returns already has strong performance. The previous results compares CQL (ReDS) to baselines in tasks where the MDP states are low-dimensional vectors. Next, we study vision-based robotic manipulation tasks.

**Visual robotic manipulation.** We consider two types of manipulation tasks. In the "Pick & Place" task, the algorithm controls a WidowX robot to grasp an object and place it into a tray located at a test

| Task & Dataset | EDAC | BEAR | CQL | IQL | INAC | RW & AW | EQL | SQL | XQL-C | Ours |
|---|---|---|---|---|---|---|---|---|---|---|
| medium-noisy | 0 | 0 | 55 | 44 | 0 | 5 | 0.0 | 0.7 | 4.3 | **73** |
| medium-biased | 0 | 0 | 73 | 48 | 0 | 0 | 6.5 | 8.0 | 11.7 | **74** |
| large-noisy | 0 | 0 | 42 | 39 | 0 | 10 | 7.1 | 2.9 | 11.3 | **53** |
| large-biased | 0 | 0 | 50 | 41 | 0 | 8 | 8.5 | 0.5 | 7.3 | 45 |

Table 2: CQL (ReDS) outperforms prior offline RL methods including methods (IQL, XQL-C), and prior support constraint methods (BEAR, EDAC, SQL, EQL, RW & AW) on three out of four scenarios when learning from heteroskedastic data in the antmaze task. The improvement over prior methods is larger when learning from the `noisy` datasets, which are more heteroskedactic, as in Table 3, compared to `biased` datasets.

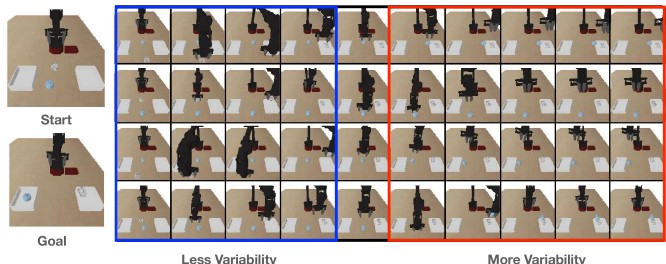

Figure 4: **Examples rollouts in the heteroskedastic bin-sort data.** In this task, an offline RL method must sort objects in front of it into two bins with a dataset that has non-uniform coverage at different states, using visual input. In the first half of the trajectory, the states exhibit a more narrow action distribution but the second half admits a more uniform action distribution.

location, directly from raw $128 \times 128 \times 3$ images and sparse 0/1 reward signal. The dataset consists of behavior from suboptimal grasping and placing policies, and the positions of the tray in the offline dataset very rarely match the target test location. The placing policies exhibit significant variability, implying these datasets are heteroskedastic under our definition. We also consider "Bin Sort" task (see Figure 4), where a WidowX robot is controlled to sort two objects into two separate bins. Here, heteroskedacity is introduced when sorting objects into the desirable bins. Similar to the Pick & Place task, the placing policy exhibits significant variability, showing an object placed in the incorrect bin (e.g., recyclable trash thrown into the non-recyclable bin). However, the grasping policy is more expert-like grasping the object with low variability. More details in Appendix E.

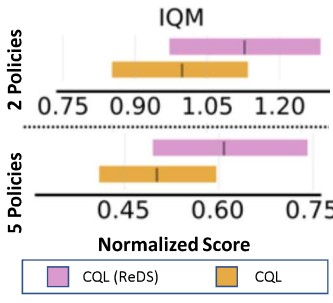

Figure 5: **CQL vs ReDS**: IQM normalized score for 10 Atari games. We consider two dataset compositions.

Table 4 presents the results on these tasks. We utilize oracle policy selection analogous to the antmaze experiments from Table 2. Table 4 shows that CQL (ReDS) outperforms CQL attaining a success rate of about 15.1% for the visual pick and place task, whereas CQL only attains 6.5% success. While performance might appear low in an absolute sense, note that both CQL and ReDS do improve over the behavior policy, which only attains a success rate of 4%. Thus offline RL does work on this task, and utilizing ReDS in conjunction with the standard distributional constraint in CQL does result in a boost in performance with this heteroskedastic dataset. For the "Bin Sorting", our method outperforms CQL by **3.5x** when learning from more heteroskedastic datasets. This indicates the effectiveness of our method in settings with higher heteroskedasticity.

| Task | | CQL | CQL (ReDS) | std $D(\pi, \pi_\beta)(\mathbf{s})$ | max $D(\pi, \pi_\beta)(\mathbf{s})$ |
|---|---|---|---|---|---|
| Pick & Place | | $6.5 \pm 0.4$ | $\mathbf{15.1 \pm 0.4}$ | 48.7 | 307.4 |
| Bin Sort (Easy) | | $\mathbf{31.2 \pm 0.3}$ | $\mathbf{31.4 \pm 0.3}$ | 7.9 | 81.6 |
| Bin Sort (Hard) | | $6.1 \pm 0.2$ | $\mathbf{23.1 \pm 0.7}$ | 59.6 | 988.3 |

Table 4: **CQL (ReDS) vs CQL** on robotic manipulation tasks. CQL (ReDS) outperforms CQL significantly when learning from more heteroskedastic datasets, as measured by $C_{\text{diff}}^\pi$: the standard deviation and the maximum of $D(\pi, \pi_\beta)(\mathbf{s})$ across states.

**Atari games.** We collect data on 10 Atari games from multiple policies that behave differently at certain states while having similar actions otherwise. We consider a case of **two** such policies, and a harder scenario of **five**. We evaluate the performance of CQL (ReDS) on the Atari games using the evaluation metrics from prior works [2, 34]. Figure 5 shows that in both testing scenarios: with the mixture of two policies (top figure) and the mixture of five policies (bottom figure), CQL (ReDS) outperforms CQL in aggregate.

**To summarize**, our results indicate that incorporating CQL (ReDS) outperforms distribution constraints with heteroskedastic datasets in a variety of domains.

## 6 Related Work

Offline Q-learning methods utilize mechanisms to prevent backing up unseen actions [39], by applying an explicit behavior constraint that forces the learned policy to be "close" to the behavior policy [23, 53, 44, 49, 53, 30, 28, 27, 52, 15], or by learning a conservative value function [33, 54, 41, 57, 56, 47, 24, 53]. Most of these offline RL methods utilize a distribution constraint, explicit (e.g., TD3+BC [15]) or implicit (e.g., CQL [33]), and our empirical analysis of representative algorithms from either family indicates that these methods struggle with heteroskedastic data, especially those methods that use an explicit constraint. Model-based methods [26, 56, 4, 51, 45, 38, 57] train value functions using dynamics models, which is orthogonal to our method.

Some prior works have also made a case for utilizing support constraints instead of distribution constraints, often via didactic examples [30, 29, 39], and devised algorithms that impose support constraints in theory, by utilizing the maximum mean discrepancy metric [30] or an asymmetric f-divergences [53] for the policy constraint [53]. Empirical results on D4RL [13] and the analysis by Wu et al. [53] suggest that support constraints are not needed, as strong distribution constraint algorithms often have strong performance. As we discussed in Sections 3.2 (Theorem 3.2 indicates that this distribution constraints may not fail when $C^\pi_{\text{diff}}$ is small, *provided these algorithms are well-tuned*.) and 4, these benchmark datasets are not heteroskedastic, as they are collected from policies that are equally wide at all states and centered on good actions (e.g., Antmaze domains in [13], control suite tasks in Gulcehre et al. [21]) and hence, do not need to modulate the distribution constraint strength. To benchmark with heteroskedastic data, we developed some novel tasks which may be of independent interest beyond this work, and find that our method ReDS can work well here.

## 7 Discussion, Future Directions, and Limitations

We studied the behavior of distribution constraint offline RL algorithms when learning from heteroskedastic datasets, a property we are likely encounter in the real world. Naïve distribution constraint algorithms can be highly ineffective in such settings both in theory and practice, as they fail to modulate the constraint strength per-state. We propose ReDS, a method to convert distributional constraints into support-based constraints via reweighting, and validate it in CQL. A limitation of ReDS is that it requires estimating the distribution $\rho_\psi$ to enforce a support constraint, which brings about its some additional compute overhead. Additionally, the instantiation of ReDS we develop in Section 4.1 is specific to methods that utilize a conservative regularizer such as CQL (or related approaches like COMBO). We clarify that our main contribution in this work is an analysis of when distributional constraints fail (which we study for AWR and CQL), and developing a principle for reformulating distributional constraints to approximate support constraints via reweighting. Devising approaches for enforcing support constraints that do not require extra machinery is a direction for future work. Understanding if support constraints are less sensitive to hyperparameters or are more amenable to model election is also a direction for future work.

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
