# OpenReview forum: "ReDS: Offline RL With Heteroskedastic Datasets via Support Constraints"
_NeurIPS.cc/2023/Conference — NeurIPS 2023 poster_

### Official Review · Reviewer_huak · 2023-06-25

**Soundness:** 3 good
**Presentation:** 3 good
**Contribution:** 2 fair
**Rating:** 5
**Confidence:** 4

**Summary:**

This paper introduces ReDS for offline reinforcement learning (RL) where the variability of demonstrated behaviours changes non-uniformly across the state space. Unlike prior offline RL methods, e.g., CQL and AWR, that directly constrain the learning policy by distribution matching with the behaviour policy, ReDS presents the task as support matching and implements it by reweighting the data distribution in CQL. Both theoretical analysis and experimental results are provided to demonstrate the effectiveness of the resulting algorithm, CQL (ReDS).

**Strengths:**

- The paper is well-written and easy to follow.
- The paper is well-motivated, theoretically sound, and supported by extensive experiments.

**Weaknesses:**

There are several main weaknesses of the paper.

- The main issue of the paper is with its novelty. The idea of reweighting samples by state-dependent weights is not novel. Specifically, [1] has proposed to reweight the samples by trajectory returns / advantages and demonstrated that such a scheme significantly improves the performance of multiple offline RL algorithms, including CQL, IQL, TD3+BC, and BC. Similarly, [2] has proposed to cast the offline RL constraints as a support matching problem, rather than distribution matching, by reweighting the data samples by normalised trajectory returns. The proposed method in [2] has also been demonstrated to improve a wide range of offline RL methods, ranging from CQL, IQL, TD3+BC to CRR. Nevertheless, despite their importance,  highly similar motivation, problem formulation, and solutions, both [1] and [2] are missing from the related works.
- Although the paper has conducted extensive experiments across different domains, it lacks several important studies. Similar methods [1, 2] have already demonstrated that support matching methods for offline RL can be algorithm-agnostic, i.e., it can be used to boost the performance of different algorithms, ranging from CQL to IQL, TD3+BC, and etc. However, this paper only discusses CQL (ReDS). From the perspective of algorithmic design, CQL, TD3+BC, and AWR (discussed by the authors in line 127) / CRR / IQL (based on the AWR for policy improvement) are all distribution-constraint algorithms. It is important to discuss how ReDS could improve the aforementioned algorithms and how ReDS differs from [1, 2] in its performance.

References:

[1] Hong, Z. W., Agrawal, P., des Combes, R. T., & Laroche, R. (2023, February). Harnessing mixed offline reinforcement learning datasets via trajectory weighting. In The Eleventh International Conference on Learning Representations.

[2] Yue, Y., Kang, B., Ma, X., Xu, Z., Huang, G., & Yan, S. (2022). Boosting Offline Reinforcement Learning via Data Rebalancing. arXiv preprint arXiv:2210.09241.

**Questions:**

- Some of the important domains / tasks are missing from the paper. As a standard benchmark for offline RL, the medium-level locomotion tasks, the adroit tasks, and the original antmaze tasks are missing from the paper. Specifically, the original antmaze itself is already heteroskedastic in my opinion, as for most of the states, the ant only needs to go straight and only occasionally it will make turns. How does CQL (ReDS) compare with the well-benchmarked results on these tasks?
- How to apply ReDS to other distribution-constraint-based methods, including AWR, TD3+BC, etc. If it can be applied, how does it improve them and how does it compare with the aforementioned related works? If ReDS cannot be applied to other algorithms, what is its limitation?

**Limitations:**

The main limitation of the paper is with its novelty. The idea of reweighting samples and casting the distribution-constrained offline RL, i.e., distribution matching problem, as a support matching problem, has been discussed by prior methods. Also, many important and well-benchmarked domains and tasks are missing in the experiments, which makes it less clear about the performance of the method. Overall, I do think this is an interesting paper, but given the current limitations and issues, it might not be ready for publication yet.

---

> ### Author Rebuttal · Authors · 2023-08-10
>
> Thank you for the thoughtful response. Regarding novelty, we would like to clarify that while prior works exist that utilize re-weighting with offline RL algorithms, in this work, we not only develop a method for re-weighting distributional constraint methods but also analyze challenges with distribution constraints on heteroskedastic datasets. That is, our contribution isn’t just in developing a re-weighting method for offline RL, but in understanding challenges with existing methods and showing how re-weighting can alleviate these challenges. To further address your concerns, we compare ReDS to the method Hong et al. 2023 and find ReDS outperforms this method, indicating its efficacy. We discuss this and other questions in detail below. **Please let us know if your questions are resolved, and if so, we would appreciate it if you could update your score.  We would be happy to discuss any further questions.**
>
> ***Comparisons with Hong et al. 2023***
>
> Thank you for pointing out these comparisons. We do believe that these methods are quite relevant and we will discuss them in the related works section of the paper. We also ran experiments on the heteroskedastic AntMaze domains to evaluate these methods and present our results below. In brief, we found that ReDS still outperforms this prior method.
>
> **RW and AW (Hong et al. 2023):** We tried two variations that the authors use for weighting: Reward Weighting (RW) and Advantage Weighting (AW), using the author’s hyperparameters for the D4RL AntMaze tasks. We compared against the best-performing variation for each dataset. For these same variations and hyperparameter settings, we are also able to replicate the author’s performance on the D4RL AntMaze tasks.
>
> Note that the performance is significantly worse compared to the base algorithm (IQL) used by these methods and our approach, CQL (ReDS). Note this isn’t contradictory:  the author’s reported numbers are outperformed by IQL (baseline; with uniform sampling) on the D4RL AntMaze tasks.
> A full comparison is found below:
> | Task & Dataset |     EDAC     |    BEAR     |    CQL    |    IQL    |    INAC    | RW and AW | ReDS [Ours] |
> |:--------------:|:------------:|:-----------:|:---------:|:---------:|:----------:|:---------:|:---------------:|
> |  medium-noisy  |      0       |      0      |    55     |    44     |     0      |     5     |      **73**     |
> | medium-biased  |      0       |      0      |  **73**   |    48     |     0      |     0     |      **74**     |
> |  large-noisy   |      0       |      0      |    42     |    39     |     0      |    10     |      **53**     |
> | large-biased   |      0       |      0      |  **50**   |    41     |     0      |     8     |        45       |
>
> We aren’t able to compare with Yue et al in time for the rebuttal but will in the final.
>
> **Why does ReDS outperform AW and RW?:** We believe this limitation stems from the re-weighting of entire trajectories. We wouldn’t expect such a strategy to help with heteroskedasticity especially when states with both narrow and wide behavior action distributions appear in the same trajectory such as in the heteroskedastic AntMaze or even in our didactic example in Figure 1. Reweighting a trajectory wouldn’t provide precise-enough control to change the behavior distribution to impose the correct state-specific re-weighting.
>
> ***As a standard benchmark for offline RL, the medium-level locomotion tasks, the adroit tasks, and the original AntMaze tasks are missing from the paper***
>
> While the standard D4RL tasks provide an effective way to benchmark offline RL methods, our focus in this work was to validate if ReDS can be effective with heteroskedastic datasets. Since D4RL tasks consist of data from policies that typically exhibit uniform action entropy across states, they tend to not be heteroskedastic as observed in Table 2a. Due to this reason, we thought it was more important to study the performance of ReDS on heteroskedastic datasets while making sure it does not degrade performance on a subset of standard D4RL tasks (as in Table 1). That said, we are running additional experiments on medium locomotion tasks and Antmaze tasks, and will report back on the status of our results before the end of the discussion period.
>
> ***Support matching methods for offline RL can be algorithm-agnostic***
>
> While we agree that support-matching methods for offline RL can be domain agnostic, in this paper, we attempted to instantiate ReDS for CQL: an instantiation specific to methods that use a conservative regularizer like CQL (or related approaches like COMBO). Here, we modify the distribution for push-up and push-down in the CQL regularizer (Figure 3), but these loss terms aren’t present in other algorithms such as policy constraints. We would like to clarify that we don’t claim that ReDS is algorithm agnostic. Other works in offline RL also propose algorithms that aren’t general methodologies (e.g., TD3+BC, IQL, etc). As such, we believe that the lack of a general methodology should not be used for rejecting the paper, provided the method attains good results.

---

> > ### Comment · Reviewer_huak · 2023-08-16
> >
> > I appreciate the authors' comprehensive rebuttal and the inclusion of additional experiments. The rebuttal effectively addresses some of my concerns. Regarding the additional experiments conducted on antmaze, I share a similar concern as Reviewer uYnF.  It looks interesting to me that CQL is achieving a fairly strong performance on these harder tasks (comparable with ReDS on medium-biased and outperforming ReDS on large-biased). Specifically, the results are even better than the reported results of CQL on the less challenging original antmaze domains. Please find below the original reported results.
> >
> > |                           |  CQL |
> > |---------------------------|:----:|
> > |   antmaze-medium-play-v0  | 61.2 |
> > | antmaze-medium-diverse-v0 | 53.7 |
> > |   antmaze-large-play-v0   | 15.8 |
> > |  antmaze-large-diverse-v0 | 14.9 |
> >
> > It makes me wonder how much of the strong performance of ReDS comes from CQL and how much it comes from the algorithmic advancements. It would be much appreciated if you could give some additional explanations. In addition, I checked the appendix and found that ReDS is developed based on the implementation of JaxCQL. I happened to use this repository before and failed to reproduce CQL's results on antmaze. Could you kindly share your full hyper-parameters?
> >
> > I'd love to increase my rating if the concern is addressed.

---

> > > ### Author Response · Authors · 2023-08-18
> > > **Response to Reviewer huak**
> > >
> > > Thank you for your reply! The full range of hyperparameters that we use for JaxCQL is shown below. Using this set of hyperparameters, we were able to reproduce the numbers for CQL on the JaxCQL repository, as other works have also done in the past: e.g., Cal-QL (Nakamoto et al. 2023) and the CQL implementation in the open-source CORL (Clean Offline RL) library (Tarasov et al. 2022): https://github.com/tinkoff-ai/CORL.
> > >
> > > |      Hyperparameters         |        Values       |
> > > |:---------------------------:|:------------------:|
> > > |        CQL Lagrange          |        True        |
> > > | CQL Lagrange Target Action Gap|        0.8         |
> > > |      Critic Network          |   256-256-256-256  |
> > > |       Actor Network          |       256-256      |
> > > |       Reward Scale           |         10         |
> > > |        Reward Bias           |         -5         |
> > > |         Critic Lr            |       3e-4        |
> > > |          Actor Lr            |       3e-4        |
> > >
> > > **Where does improvement in ReDS come from?** Of course, since ReDS builds directly on top of CQL, its performance will be heavily affected by the performance of the CQL algorithm. So we would expect improvements in design decisions for CQL to improve the performance of ReDS as well. That said, please note that across our experimental results, we do observe a significant improvement with using the weighting proposed by ReDS: for example, in Table 3 on the visual manipulation tasks and in Figure 5, on the Atari tasks, as well as the noisy antmaze tasks. This improvement from ReDS is not explained by baseline CQL as we utilize the same hyperparameters for ReDS, and hence the gains can only be attributed to the weighting prescribed by ReDS.
> > >
> > > **CQL performance on AntMaze:** With regards to the performance of CQL on antmaze tasks, note that while the original CQL paper studies the antmaze-v0 tasks, D4RL deprecated these tasks in 2021 and upgraded to antmaze-v2 datasets, which fixed inconsistencies between terminals and rewards in the antmaze-v0 datasets (https://github.com/Farama-Foundation/D4RL/commit/49699950ae0c00501114b420626e956c0437d00e). We suspect this is the reason for performance improvement for the baseline CQL from reported v0 to our v2 results. To address your point about results on the heteroskedastic datasets, we note that on the D4RL medium antmaze datasets, CQL attains a larger performance than what it attains on our -biased and -noisy datasets. Finally, as we also note in Lines 364-365 in the submission, our results used oracle policy selection for reporting results for all methods, following trends in recent works such as ATAC (Cheng et al. ICML 2022; best paper runner-up)  due to a lack of an early stopping algorithm for offline RL methods. However, D4RL reports last iterate results, which also partly explains a smaller gap in performance on the heteroskedastic antmazes and the D4RL antmaze. We will add a detailed discussion in the updated paper.
> > >
> > > **We would be grateful if you are willing to raise your score if your concerns are addressed. We would be happy to discuss further if you have any remaining questions.**

---

> > > > ### Comment · Reviewer_huak · 2023-08-20
> > > >
> > > > I thank the authors for the detailed expalanations. They have addressed some of my concerns. I'll slightly increase my rating.

---

### Official Review · Reviewer_5pSd · 2023-07-03

**Soundness:** 4 excellent
**Presentation:** 4 excellent
**Contribution:** 3 good
**Rating:** 6
**Confidence:** 4

**Summary:**

The paper has identified the presence of heteroskedasticity in realistic offline RL datasets, which negatively impacts the performance of existing offline RL methods that rely on distributional constraints. To tackle this issue, the authors introduce a new approach called ReDS, which transforms distributional constraints into support-based constraints.  Several novel heteroskedastic datasets were introduced in order to showcase the superior performance of ReDS compared to existing offline RL methods.

To practically achieve the support constraint, the fundamental change based on CQL is to add a new penalty term $\pi_{re}$, which can be obtained via
$\arg \max_{\rho_\psi} \mathbb{E}_{s \sim \mathcal{D}, a \sim \pi_\beta(\cdot \mid s)}[\log \rho_\psi(a \mid s) \cdot \exp (-A_\theta(s, a) / \tau)]$.

**Strengths:**

1. Well-written, with high-quality examples, and technically sound.
2. Extensive evaluations over many different tasks.

**Weaknesses:**

1. The design of the reweighted version of the behavior policy in Eq. 6 is heuristic. I can not see a clear clue why the coefficient is designed as 1/2 and 1/2.
2. The experiments in the noisy and biased version of Antmaze tasks seem contrived.
3. Inconsistent of the differential concentrability between theory and real-world practical datasets (std).

**Questions:**

1. Could the policy $\pi$ in Deﬁnition 3.1 be any policy? What if it is an arbitrary worst policy, i.e., random policy? I do not see clear evidence that $C_{diff}$ is big enough when $D(\pi, \pi_{\beta})$ is arbitrarily large in both $s_1$ and $s_2$. On the other hand, should $s_1, s_2 \sim d^{\pi}$ without any other condition? What if they are all sampled from the same place, i.e., the narrow hallways in the didactic example?
2. Any detail about how to compute the std in Chapter 4.2?
3. How well does ReDS perform over non-heteroskedastic datasets (random, medium, expert)? Is there any limitation to on non-heteroskedastic datasets?

---

> ### Author Rebuttal · Authors · 2023-08-10
>
> Thanks for the feedback and positive assessment of our work.  We address your concerns below and will update the paper to clarify each of the questions. Please let us know if your questions are resolved, we are happy to discuss further if any questions are remaining.
>
> ***Inconsistent of the differential concentrability between theory and real-world practical datasets (std).***
>
> The intuitive explanation for what makes a dataset heteroskedastic -- namely, that the variability in the policy is different in different states -- is provided primarily to build intuition. Unfortunately, this notion by itself does not make a great formal definition, because it does not capture the count of states $n(s)$ that shows up in a safe-policy improvement bound (the constraint in Equation 5). Therefore, in our analysis, we use the somewhat more complex notion of differential concentrability, which captures a similar intuition (we make the mathematical connection between the intuition of heteroskedasticity and our definition precise below) but provides us with the foundation on which to build our theoretical analysis.
>
> To better understand the relationship between the variance of the action distribution and our Definition 3.1, consider a simpler scenario where we remove the counts $n(s)$ from the expression of differential concentrability and set $\pi$ in $C^\pi_\text{diff}$ to be the uniform distribution over actions. Then, we can show that the value of $C^\pi_\text{diff}$ is **exactly** equal to twice the variance of entropy of the dataset action distribution across states, which matches with the metric (variance = square of the standard deviation) we measure in our experiments.
>
> Of course, we cannot exclude the counts of states $n(s)$ for technical accuracy, however,  it should be noted that in high-dimensional state space tasks, such as those examined in our experiments, each state in the offline dataset is likely to be unique, thus validating the assumption that $n(s) = 1$. This means that measuring the variance in the entropy of the action distribution is an accurate estimate of differential concentrability in our experiments.
>
> ***Could the policy $\pi$  in Deﬁnition 3.1 be any policy?***
>
> While the choice of the policy $\pi$ in the _expression for_  differential concentrability can be any policy, we would like to clarify that we utilize the value of $C^\pi_\text{diff}$ only for the policy learned via distributional constraint methods to define how heteroskedastic a dataset is. This is akin to the definition of standard concentrability which, in principle, can be computed for any policy but only its value at the optimal policy is a useful measure of the hardness of the offline RL problem.
>
> **I do not see clear evidence that $C^\pi_\text{diff}$ is big enough when $D(\pi, \pi_\beta)$ is arbitrarily large in both $s_1$  and $s_2$. **
>
> We would like to clarify that to the best of our knowledge, we did not claim this in the paper. In fact, in lines 163-168, we claim otherwise: we say that $C^\pi_\text{diff}$ can be _small_ even when $D(\pi, \pi_\beta)$ is large enough because it might take similar values at different states. Does this clarify your question?
>
> **Should $s_1, s_2 \sim d^\pi$  without any other condition? What if they are all sampled from the same place?**
>
> Please note that $d^\pi$ refers to the state occupancy distribution of the policy that is attempting to maximize the conservative offline RL objective (Equation 1). Intuitively, this means that sampling $s_1$ and $s_2$ from $d^\pi$ should be enough because $d^\pi$ is, in effect, close to $d^{\pi_\beta}$. Thus, when the dataset behavior distribution has a lot of variability across states, this variability would also be reflected on states sampled from $d^\pi$.
>
> ***Any detail about how to compute the std in Chapter 4.2?***
>
> We apologize, but we are not sure which table this question is referring to. Could you please provide a clarification? If you are referring to Table 2, the standard deviation is computed by first recording the value of KL divergence between a model of the behavior policy (trained by maximizing the log-likelihood of the dataset) and the policy learned by CQL for the best hyperparameter $\alpha$ for all states in the dataset, and then, computing the standard deviation in this divergence value across all states in the dataset.
>
> ***Why the coefficient is designed as $\frac{1}{2}$ and $\frac{1}{2}$.***
>
> Setting this coefficient to a value of $\frac{1}{2}$ was the initial choice for our experiments. We found this choice was convenient: it worked well in practice and circumvented the need for introducing a new hyperparameter. That said, future work could aim to study the sensitivity of ReDS to other values of this coefficient, which we believe will only improve the performance of this approach.
>
> ***How well does ReDS perform over non-heteroskedastic datasets (random, medium, expert)? Is there any limitation to non-heteroskedastic datasets?***
>
> We are running these experiments and will get back to you before the end of the author-reviewer discussion period. However, note that we do already have experiments on some standard and commonly used non-heteroskedastic benchmark tasks in Table 1.

---

> > ### Comment · Reviewer_5pSd · 2023-08-18
> > **Official Comment by Reviewer 5pSd**
> >
> > I appreciate the authors' response to my previous comments. Concerning the standard deviation, I was referring to Table 2. I apologize for the ambiguity in my previous question. Most of my concerns have been addressed.
> >
> > However, I still have a few points that require further clarification in Definition 3.1. To clarify my question: I believe that $C_{\text{diff}}^{\pi}$ would be \textbf{small} when the learned policy $\pi$ is arbitrarily worse (i.e., a random policy), because both $D(\pi, \pi_{\beta})(s_1)$ and $D(\pi, \pi_{\beta})(s_2)$ would be large enough to make $C_{\text{diff}}^{\pi}$ small.
> >
> > Additional Questions:
> >
> > I have also reviewed comments from other reviewers. Reviewer uYnF mentioned advanced methods within the in-sample learning paradigm based on SAC, such as InAC. It would enhance the completeness of this paper if experiments and analysis of these advanced in-sample methods [1, 2] were conducted. For instance, SQL (Xu et al., 2023, Chapter 5.2) demonstrates that sparsity in value function learning can be beneficial in noisy datasets (heteroskedastic datasets) and outperform IQL.
> >
> > [1] Xu H, Jiang L, Li J, et al. Offline rl with no ood actions: In-sample learning via implicit value regularization[J]. ICLR, 2023.
> > [2] Garg D, Hejna J, Geist M, et al. Extreme q-learning: Maxent RL without entropy[J]. ICLR, 2023.

---

> > > ### Author Response · Authors · 2023-08-18
> > > **Response to Reviewer 5pSd**
> > >
> > > Thank you for your prompt response! We are glad that most of your concerns are addressed. Regarding your question about the definition of $C^\pi_\text{diff}$: you are right that this value would be small when $\pi$ is a random policy such that $D(\pi, \pi_\beta)(s)$ is large. However, please note that to decide when a given offline RL problem is heteroskedastic and, more importantly, to identify when  distribution constraints algorithms will fail, Theorem 3.2 only considers the value of $C^\pi_\text{diff}$ for $\pi := \pi_\alpha$ (i.e., policies obtained by optimizing the generic offline RL objective in Equation 1) and not just any policy $\pi$. Below, we enumerate and intuitively discuss various cases in which a generic offline RL objective would produce a random policy at all states and reason as to why in these settings a low value of $C^\pi_\text{diff}$ is still consistent with the behavior of distribution constraint algorithms. There are two possible scenarios under which the learned policy can be a random policy:
> > >
> > > 1. **The behavior policy is highly random at all states as well**, in which case the offline RL problem is not heteroskedastic because the variability in the behavior distribution is similar across all states (and thus, a small value of $C^\pi_\text{diff}$ correctly identifies this problem as non-heteroskedastic). In this case, we can improve the learned policy performance by tuning the value of $\alpha$ in Equation 1.
> > >
> > > 2. **When the behavior policy action distribution is not random, but it does not cover any action that maximizes reward** (and hence optimizing Equation 1 recovers a high-entropy policy as it attempts to maximize the sum of two conflicting objectives: reward and divergence against the behavior policy). For concrete understanding, consider a setting when the reward is binary, and no actions in the dataset attain a +1 reward. This problem is simply a hard offline RL problem as no (near-)optimal behavior is covered by the offline dataset. It is not necessarily heteroskedastic, as we would not expect support constraint algorithms to offer a distinct benefit over distribution constraint algorithms, all of which will perform poorly. This is because the behavior policy does not cover good actions in any state at all and all actions inside the support attain poor performance. Hence, a small value of $C^\pi_\text{diff}$ also describes the heteroskedasticity challenges associated with this scenario, by triviality.
> > >
> > > Hence, when considering only policies that maximize the generic offline RL objective, $C^\pi_\text{diff}$ will be large only when distribution constraint algorithms for any $\alpha$ fail because a single $\alpha$ is insufficient to modulate the strength of the behavior regularization at all states.
> > >
> > > **Regarding your additional question,** Thank you for the suggestion! We are going to run experiments with these in-sample learning methods as well and add them to the final version of the paper.

---

> > > > ### Comment · Reviewer_5pSd · 2023-08-18
> > > > **Thanks for the Authors Repsonse**
> > > >
> > > > I appreciate the authors' response to my concerns about Definition 3.1. Due to the heuristic design of the coefficient, I will maintain my current rating. I understand that it is somewhat late in the discussion period, but I would be willing to raise my rating in this period if experiments and analyses regarding my additional questions are performed. I believe this should be relatively straightforward to implement, as the relevant code (SQL/EQL) is open-sourced and the dataset is easy to construct. I look forward to seeing your insightful analysis and comparisons.

---

> > > > > ### Author Response · Authors · 2023-08-20
> > > > > **Thank you and response to your comments**
> > > > >
> > > > > Thank you for your request and for engaging with us. We ran the 3 methods you have mentioned on the heteroskedastic antmaze datasets: SQL, EQL and XQL-C. We used the author’s implementation and utilized the recommended hyperparameters for the antmaze domains on these tasks. We ran 3 seeds on each task for reproducibility for 1M steps. Below is a full table for comparison.
> > > > >
> > > > > | Dataset | EDAC | BEAR |  CQL  |  IQL  |  INAC  | RW and AW | ReDS [Ours] |  EQL  |  SQL  | XQL-C |
> > > > > |:--------------:|:----:|:----:|:-----:|:-----:|:------:|:---------:|:-----------:|:-----:|:-----:|:-----:|
> > > > > |  medium-noisy  |  0   |  0   |   55  |   44  |   0    |    5      |    **73**   |  0.0  |  0.7  |  4.3  |
> > > > > | medium-biased  |  0   |  0   | **73**|   48  |   0    |    0      |    **74**   |  6.5  |  8.0  | 11.7  |
> > > > > |  large-noisy   |  0   |  0   |   42  |   39  |   0    |   10      |    **53**   |  7.1  |  2.9  | 11.3  |
> > > > > | large-biased   |  0   |  0   | **50**|   41  |   0    |    8      |      45     |  8.5  |  0.5  |  7.3  |
> > > > >
> > > > > Note that these in-sample methods perform better than the in sample method of INAC but do not outperform our approach and some other baseline methods.
> > > > >
> > > > > We are happy to also evaluate these methods on the other domains in the final version of the paper. Please let us know if your concerns are addressed.

---

> > > > > > ### Comment · Reviewer_5pSd · 2023-08-21
> > > > > > **Thanks for the Experiments from Authors**
> > > > > >
> > > > > > This is a good paper! I am grateful for the authors' efforts in conducting additional experiments on in-sample methods. Having carefully revisited the comments from other reviewers and my previous concerns, I have decided to maintain my current rating.

---

### Official Review · Reviewer_uYnF · 2023-07-06

**Soundness:** 3 good
**Presentation:** 3 good
**Contribution:** 3 good
**Rating:** 6
**Confidence:** 5

**Summary:**

The paper introduces ReDS, a novel offline RL method designed to handle heteroskedastic datasets. ReDS incorporates support constraints by reweighting the data distribution based on conservative Q-learning (CQL). This allows the learned policy to deviate from the behavior policy within its support while maximizing long-term return.

**Strengths:**

1. The addressed problem of a more fine-grained support constraint in offline RL research is crucial.
2. The paper includes thorough discussions about the proposed method.
3. The introduction of the method, which utilizes reweighting distribution constraints to achieve a fine-grained support constraint, is clear and reasonable. The explanation in Figure 3 is particularly intuitive.
4. The method demonstrates significant performance advantages, especially when applied to heterogeneous datasets.

**Weaknesses:**

1. The didactic example does not convincingly support the claims. For example, it does not convincingly demonstrate the characteristics of heterogeneous datasets in key states (such as the bottom-left corner of the map) and their impact on the CQL and REDS algorithms. Additionally, the results presented in Figure 1 show that the baseline algorithms AWR and CQL fail, while REDS does not. However, has this demonstration accounted for the potential impact of statistical errors?
2. The baseline algorithm (CQL) seems somewhat outdated. It would be necessary to discuss and compare with more advanced methods that satisfy the support constraint, such as InAC[1].
3. There are several spelling errors throughout the paper. A careful proofreading is necessary. For instance, the capitalized "w" in "With" in the title should be lowercase, and the occurrence of "WSome" in line 419 is incorrect.

[1] Xiao, Chenjun, et al. "The in-sample softmax for offline reinforcement learning." arXiv preprint arXiv:2302.14372 (2023).

**Questions:**

The main concern relates to the points mentioned in the weaknesses section. Additionally, there is another question that requires clarification:

Q: Is the proposed method plug-and-play? If so, why not integrate it with an updated or foundational algorithm (e.g., SAC+vanilla distribution constraint)?

**Limitations:**

The paper discusses the limitations. And, these limitations seems not appear to require immediate attention.

---

> ### Author Rebuttal · Authors · 2023-08-10
>
> Thank you for the feedback and positive assessment of our work.  To address your concerns, we add 2 comparisons to the support-constraint methods InAC and Hong et al 2023. We also explain how we took statistical errors into account when we compute the statistics for the didactic example.
>
> **Please let us know if our answers resolve your concerns, and if so, we would truly appreciate it if you are willing to upgrade your assessment. We are happy to discuss any further questions.**
> ___
>
> ***Additional baselines; CQL is outdated***
>
> We add 2 additional baselines and compare our method ReDS to InAC and Hong et al. 2023 on the heteroskedastic AntMaze datasets.
>
> **InAC [Xiao et al. 2023]** We find this approach fails to attain a non-zero performance on the heteroskedastic AntMaze tasks.  We tuned InAC by sweeping the learning rate and temperature $\tau$ over the following values:
>
> |   Hyperparameters   |       Values       |
> |:-------------------:|:-----------------:|
> |        Tau          | 0.5, 0.33, 0.1, 0.01 |
> |   Learning Rate     | 1e-3, 3e-4, 1e-4, 3e-5 |
>
> For each sweep value, we are unable to get non-zero performance. We also ran this method on the D4RL AntMaze tasks, unable to attain non-zero performance even on these standard tasks. Note, our findings don’t contradict the results in InAC because the paper doesn’t study them. This indicates that ReDS outperforms InAC on the heteroskedastic AntMaze tasks.
>
> **Harnessing Mixed Offline Reinforcement Learning Datasets via Trajectory Weighting [Hong et al. 2023]**: We see that performance for the AntMaze heteroskedastic tasks using the hyperparameters the authors used for the AntMaze results in their paper, is non-zero but lower than the performance of ReDS. We tried two variations that the authors recommended for weighting: Reward Weighting (RW) and Advantage Weighting (AW). We compared against the best-performing variation for each dataset. For these same hyperparameters, we verify performance on the D4RL AntMaze tasks, which match the author’s results.
>
> For both algorithms, we used the author’s implementation, averaging over 4 seeds and using the max performance after 1M training steps.
>
> Additionally, in the paper, we compare against two prior methods, BEAR and EDAC, which enforce support constraints, and find that ReDS outperforms all of these prior methods. We are happy to add more comparisons if the reviewer has any suggestions for the final version of the paper.
>
> A full comparison can be found below:
> | Task & Dataset |     EDAC     |    BEAR     |    CQL    |    IQL    |    INAC    | RW and AW | ReDS [Ours] |
> |:--------------:|:------------:|:-----------:|:---------:|:---------:|:----------:|:---------:|:---------------:|
> |  medium-noisy  |      0       |      0      |    55     |    44     |     0      |     5     |      **73**     |
> | medium-biased  |      0       |      0      |  **73**   |    48     |     0      |     0     |      **74**     |
> |  large-noisy   |      0       |      0      |    42     |    39     |     0      |    10     |      **53**     |
> | large-biased   |      0       |      0      |  **50**   |    41     |     0      |     8     |        45       |
>
> ***Didactic example doesn’t convincingly demonstrate the characteristics of heterogeneous datasets in key states.***
> The didactic gridworld consists of narrow hallways and wide rooms that periodically repeat throughout the path the policy needs to take to successfully solve the task. While in Figure 1(a), we visualize the action distributions at locations A and B only, a similar variability arises in behaviors in the bottom-left corner of the map. In regards to why AWR and CQL get stuck at the bottom left corner, this is because while these policies are still able to escape the first two wide rooms along the path, they are unable to do so the third time at the bottom of the maze. This is not due to the particular nature of that location, but the general periodic pattern of the behavior distribution. We are happy to edit this visualization for further clarity if you have any suggestions.
>
> **However, has this demonstration accounted for the potential impact of statistical errors?**
> Yes, we do account for statistical errors: **(1)** the offline dataset provided to all algorithms is of finite size, indicating that these algorithms including ReDS must still learn in the face of statistical error, and  **(2)** For visualizing how the policies behave upon evaluation, we directly plot the discounted, long-term state-action visitation following the visualizations in Fu et al. 2019, rather than plotting the empirical rollout of each algorithm, so our visualizations do take into account the impact of statistical errors. Note that the (discounted) visitation distribution of a policy is computed using T steps of forward propagation.
>
> ***Is the proposed method plug-and-play?***
> As stated in Lines 210-224,  the principle of combining re-weighting with distributional constraints is plug-and-play, applicable to any distributional constraint algorithm including SAC + distributional constraint methods. However, the instantiation of ReDS we develop in Section 4.1 is specific to methods that utilize a conservative regularizer such as CQL (or related approaches like COMBO). This is because our current instantiation of ReDS modifies the distribution for push-up and push-down in the CQL regularizer (Figure 3), but these loss terms are not present in policy constraints.  We clarify that our main contribution in this work is an analysis of when distributional constraints fail (which we study for AWR and CQL), and developing a principle for reformulating distributional constraints to approximate support constraints via reweighting. We validate this through CQL and leave as future work the application of this framework to other offline RL algorithms.
>
> ***Careful proofreading is necessary*** Thank you for the suggestion! We will thoroughly proofread to ensure the final paper is polished.

---

> > ### Comment · Reviewer_uYnF · 2023-08-15
> >
> > Thanks for your reply. Most of my concerns have been addressed. However, it's quite strange that InAC cannot achieve non-zero performance while IQL can even on original Antmaze datasets.  I would love to raise my score if you could give more analysis.

---

> > > ### Author Response · Authors · 2023-08-18
> > > **Response to Reviewer uYnF**
> > >
> > > Thank you for your reply! We are glad that most of your concerns have been addressed. With regards to InAC, please note that the original InAC paper did not study these datasets and we were unable to find any other paper  that evaluated the InAC method on the D4RL AntMaze datasets. In fact, one of the very recent papers which came out on arXiv after the NeurIPS deadline – In-Sample Policy iteration (Hu et al. 2023) – incorporated InAC as a baseline, but did not evaluate it on the medium and large AntMaze datasets (observe that Table 1 in https://arxiv.org/pdf/2306.05726.pdf reports a “-” for InAC on AntMazes).
> > >
> > > **Experiments:** We ran a sweep over some hyperparameters of InAC: temperature $\tau$ and the learning rate for the critic, using the InAC’s recommended hyperparameters and implementation from Appendix B in their paper.  All other details such as the network architecture are standard to these environments, consistent with other baselines such as CQL, IQL as well as our method. In particular, we utilize the following set of hyperparameters:
> > >
> > > |   Hyperparameters   |       Values       |
> > > |:-------------------:|:-----------------:|
> > > |        Tau          | 0.5, 0.33, 0.1, 0.01 |
> > > |   Learning Rate     | 1e-3, 3e-4, 1e-4, 3e-5 |
> > > |   Critic Network    | 256-256-256-256 |
> > > |   Actor Network     | 256-256 |
> > >
> > > We also ran an additional experiment where we shift the rewards in the dataset to -1, 0 following IQL which found these values to do better on the AntMaze domain, using the same sweep parameters. But we were still unable to attain nonzero performance for InAC on these tasks.
> > >
> > > **Empirical analysis:** In all of our runs, we found that the Q-values were heavily over-estimated: for example, we observed that the average Q-values and the value function in the dataset were always positive, although the rewards were -1 and 0 (e.g., a value of +254 for $\tau$=0.5 and learning rate of 0.01). We hypothesized that this was because the $\log \pi_\psi(a|s)$ in InAC (Equation 15) was negative and had a large magnitude. Subtracting this log probability  prevented the value backup from focusing on the reward signal and hence the policy did not learn effective behavior. In further support of our hypothesis is the fact that CQL and IQL do not backup the entropy term ($-\log \pi(a|s)$) utilized in SAC backups in their implementations.
> > >
> > > To verify this hypothesis, we ran InAC by removing this log probability term from the value function training objective, and found that the learned value function attained values that were over-estimated less, and were able to find some settings of the hyperparameters (e.g., learning rate of 0.1 and tau of 0.5) where the policy attained a non-zero return of around 20%. However, note that doing this is technically incorrect according to the theoretical derivation in InAC, in the sense that it does not actually produce an in-sample optimal Q-function. That said, our analysis localizes this as one of the issues with InAC.
> > >
> > > **We would be grateful if you would be willing to raise your score in the light of the above clarifications and analysis. Please let us know if you have any more questions. Thank you so much!**

---

> > > > ### Comment · Reviewer_uYnF · 2023-08-18
> > > >
> > > > Thanks for your effort! I have raised my score to 6. Wishing you all the best with your publication.

---

### Official Review · Reviewer_XMxS · 2023-07-06

**Soundness:** 3 good
**Presentation:** 3 good
**Contribution:** 3 good
**Rating:** 5
**Confidence:** 4

**Summary:**

The paper focuses on heteroskedastic datasets, where the distribution of actions may not be uniform in certain states but close to uniform in other states. The authors propose the ReDS method, which re-weights the actions to penalize "bad" actions that perform poorly on the dataset but have a higher probability, while encouraging "good" actions with lower probabilities in the dataset's support. ReDS outperforms previous baseline methods in multiple experimental scenarios.

**Strengths:**

The majority of the paper is written in a clear and easy-to-follow manner. The research problem addressed in this paper is indeed relevant to real-world scenarios, and the authors provide didactic examples to illustrate the potential issues with previous methods. The paper offers a detailed analysis and proposes a concise and intuitive plug-in solution called ReDS, while also demonstrating its effectiveness. The experimental settings are diverse, covering multiple domains.

**Weaknesses:**

ReDS should apply in broader scenarios beyond the specific setting of heteroskedastic datasets. Another point is the choice of experimental settings, as most of them involve manually constructed datasets to satisfy heteroskedastic conditions, rather than utilizing more intuitive scenarios such as autonomous driving or validating whether some existing datasets are really heteroskedastic. Thus the significance is somewhat limited.




Minor issues:

The figures in the main text, such as Figures 1 and 2, rely heavily on the captions. Adding more conclusive statements in the main text could enhance consistency and clarity.

Table 4 --> Table 2?

Line 326 "some of the standard D4RL datasets which are not heteroskedastic in and find" --> extra "in"

Line 419 "WSome prior works ..."

**Questions:**

1. In Fig.1, I wonder if the datasets cover the whole room B or only a point shown in the figure. If the datasets cover the whole room, AWR and CQL should be able to learn to exit room B, since other actions lead to lower returns.
2. As Line 326-327 stated, the authors have evaluated ReDS on some of the standard D4RL datasets which are not heteroskedastic and find that the addition of ReDS, as expected, does not help, or hurt on those tasks. Why does ReDS hurt the performance on some of the standard D4RL datasets? By Theorem 1, ReDS can at least have better performance.
3. Why do the experimental domains contain Atari?

**Limitations:**

The authors generated new datasets and did not conduct a universal analysis on whether heteroskedastic properties exist in some datasets. It would be beneficial to first employ a suitable method to effectively verify the presence of heteroskedastic properties in commonly used datasets before applying ReDS.

---

> ### Author Rebuttal · Authors · 2023-08-10
>
> Thank you for your feedback and positive assessment of our work.  We address your concerns below and will update the paper to clarify each of the questions. **Please let us know if your questions are resolved, and if so, we would appreciate it if you are willing to upgrade your score.  We are happy to discuss further if you have any remaining questions.**
>
> ***ReDS should apply in broader scenarios beyond the specific setting of heteroskedastic datasets***
>
> When learning from datasets that are non-heteroskedastic, ReDS perform similarly to distribution constraint algorithms, which indicates that ReDS can be used in other scenarios that distribution constraint algorithms can be used in. We don’t believe that this is a weakness of the method – the aim is not to beat all prior methods in all settings but to address a specific challenge in offline RL. We believe that the paper is scoped carefully around this claim, but we would be happy to revise it if you believe this is unclear.
>
> **The choice of experimental settings, … the significance is somewhat limited.**
>
> While we agree that the choice of experimental settings that we study involves specifically constructed heteroskedastic datasets, we believe that this is an important first step in verifying if developing support constraint methods can help in scenarios with heteroskedasticity. The nature of this contribution is akin to prior papers such as Hong et al. ICLR 2023, which proposes methods for re-weighting offline data to deal with imbalanced datasets, but evaluates only on manually constructed datasets that add noise or mix two existing D4RL datasets. We believe that this sort of evaluation is still of value to the community as it provides a stress test for existing algorithms and provides a proof of concept for new ideas. Many of the challenges present in real-world RL problems are not well reflected in existing benchmarks such as D4RL. That being said, we agree with you that the next step in this line of work is to validate our methods on real-world datasets in autonomous driving or robotics, which are heteroskedastic. We discuss some of these scenarios intuitively in Section 3 and in the introduction.  Finally, we also remark that our contribution in this paper is not just a novel algorithm, but also an analysis of why heteroskedasticity is hard for offline RL methods and understanding when support constraint methods can help. We believe that this in and of itself is valuable to the community developing offline RL algorithms.
>
> ***I wonder if the datasets cover the whole room B or only a point shown in the figure … AWR and CQL should be able to learn to exit room B since other actions lead to lower returns.***
>
> We clarify that the dataset consists of several sampled state-action transitions inside the wide room. Note that not every state-and-action pair from within this room is observed in the dataset, but rather only a subset of these samples are observed. We only highlight one single point B in Figure 1(a) for the compactness of visualization.
>
> More concretely, the dataset consists of sampled state-action-reward-next state transitions that are a subset of the cross-product of the state and action spaces. The samples are not uniformly distributed and are instead drawn from **a skewed action distribution which crucially attempts to keep the agent within the room**, which hinders the ability of algorithms such as AWR and CQL from learning how to exit wide rooms because a strong distributional constraint will make the policy closely match this skewed behavior distribution. This can be seen in the left part of Figure one, where in the narrow regions, the data consists of state and action pairs that uniformly move the agent towards the goal but in the wide rooms, the data consists of state and action pairs that skews the agent towards the edges of the room. Therefore, even if more than one transition is sampled from inside the wide room, the learned policy obtained from distributional constraint algorithms with a strong constraint will have the affinity to stay within the room and fail at the task.
>
> ***Authors have evaluated ReDS on some of the standard D4RL datasets. Why does ReDS hurt the performance of some of the standard D4RL datasets?***
>
> Observe in Table 1 that ReDS performs similarly or better than CQL for all tasks of the D4RL datasets presented except for the half cheetah medium expert task. We believe that this difference is small (only 2.1 performance points) and hence not statistically significant on this task.
>
> ***Why do the experimental domains contain Atari?***
>
> We chose Atari as a representative domain with discrete action spaces to study some of the heteroskedasticity challenges. Many offline RL papers (Badia et al. 2020, Kumar et al 2022) that we are aware of study the Atari domain using the offline datasets generated by Agarwal et al. ICML 2020. Atari also requires learnings from raw high dimensional pixel space, which adds to the breadth of the experiments in the paper since the D4RL  benchmarks only require learning from low dimensional state space.

---

> > ### Comment · Reviewer_XMxS · 2023-08-15
> >
> > Thanks for the reply. ReDS is a re-weight-style module that could apply in a broader scope, though this is not a drawback of the method itself. I still think the experiments should contain more strongly-related datasets since heteroskedastic datasets exist but do not prevail. As there is no free lunch (each offline algorithm has the most suitable task), it would be better if the authors could provide more domains that contain heteroskedastic data or metrics to judge whether the dataset is heteroskedastic so that we can figure out what kind of tasks really suit ReDS.

---

> > > ### Author Response · Authors · 2023-08-18
> > > **Response to Reviewer XMxS**
> > >
> > > Thank you for your reply and for engaging with us! We would like to note that we do provide a metric to judge heteroskedasticity in our experiments. The intuitive explanation for what makes a dataset heteroskedastic is that the variability in the behavior policy’s action distribution is different in different states. In the submission, we formalized this intuition by **developing the metric of differential concentrability** (Definition 3.1), such that a given offline RL problem and dataset is regarded as heteroskedastic if this measure of differential concentrability is high, and not otherwise.
> > >
> > > **Practical metric to judge heteroskedasticity:** While this formal definition requires the count of states $n(s)$, which we do not have in high-dimensional continuous state spaces, we presented an approximation of this metric in our experiments that practitioners can utilize for judging whether a given offline dataset is heteroskedastic. As shown in **Table 2a**, this metric can be computed by first running standard offline RL methods such as CQL, and then looking at the standard deviation in the value of $f(s) = D(\pi_\text{CQL}(\cdot|s), \pi_\beta(\cdot|s))$ across states. In our experiments, we found this metric to be predictive of heteroskedasticity – this metric took a significantly higher value for datasets that we developed in our experiments compared to standard D4RL datasets, and we found ReDS improved performance in such scenarios.
> > >
> > > **More domains:** We are happy to experiment with more domains with heteroskedastic data in the final version of the paper. Since we already include studies on D4RL-like tasks, Atari tasks, image-based manipulation domains, and some gridworld domains, we wanted to request your suggestions on which domains would be the most helpful to add for the final.
> > >
> > > Please let us know if this response addresses your concerns. We are happy to discuss more if you have any concerns remaining. Thank you so much!

---

> > > > ### Comment · Reviewer_XMxS · 2023-08-18
> > > >
> > > > Thanks for further clarification. Personally I think Atari had better be replaced with other domains which potentially have heteroskedasticity, e.g., autonomous driving datasets. If the aim is to solve a specific issue rather than proposing a very general approach like the PPO algorithm, I think it would be better to validate the performance on the most related tasks.

---

### Decision · Program_Chairs · 2023-09-21

**Decision:**

Accept (poster)

**Comment:**

The paper introduces ReDS, a new offline RL method for handling heteroskedastic datasets. ReDS incorporates support constraints by reweighting the data distribution based on conservative Q-learning (CQL). This allows the learned policy to deviate from the behavior policy within its support while maximizing long-term return. Various novel heteroskedastic datasets were introduced to showcase the superior performance of ReDS compared to existing offline RL methods. Both theoretical analyses and experimental results are provided to demonstrate the effectiveness of the resulting algorithm.

The reviewers acknowledged that most of this paper is well-written and easy to follow. They argued that the research problem addressed in this paper is relevant to real-world scenarios and that the authors provided interesting didactic examples to illustrate the potential issues with previous methods. Another positive aspect of this work is how the paper proposed a concise and intuitive plug-in solution (ReDS) while demonstrating its effectiveness. Finally, the experimental settings are diverse and cover multiple domains.

During the discussion between reviewers, a few concerns were raised. First, a reviewer argued that the Atari benchmark should be replaced with other domains that potentially have heteroskedasticity, such as autonomous driving datasets. Another reviewer brought up three concerns in particular: **(1)** There appears to be a gap between the theoretical analysis and the practical implementation of the algorithm, particularly concerning the heuristic design of the coefficient; **(2)** The paper lacks a thorough analysis explaining why ReDS outperforms in-sample learning methods; the reviewer believes that providing a more detailed analysis could strengthen this paper; and **(3)** The paper seems somewhat outdated. Finally, two other reviewers argued that parts of the paper only introduce seemingly incremental contributions, and that the experiments in the noisy and biased version of Antmaze tasks seem contrived.

Having said that, all reviewers acknowledged this work's contributions while emphasizing points that need improvement before publication. They encouraged the authors to explore these points and address the limitations brought up in the reviews when preparing an updated version of this paper.